# Silver Nanoparticles: Review of Antiviral Properties, Mechanism of Action and Applications

**DOI:** 10.3390/microorganisms11030629

**Published:** 2023-02-28

**Authors:** Angelica Luceri, Rachele Francese, David Lembo, Monica Ferraris, Cristina Balagna

**Affiliations:** 1Department of Applied Science and Technology, Politecnico di Torino, 10129 Turin, Italy; 2Laboratory of Molecular Virology and Antiviral Research, Department of Clinical and Biological Sciences, University of Turin, S. Luigi Gonzaga Hospital, 10043 Turin, Italy

**Keywords:** silver, silver nanoparticles, antiviral, antiviral mechanism, virucidal, applications

## Abstract

New antiviral drugs and new preventive antiviral strategies are a target of intense scientific interest. Thanks to their peculiar properties, nanomaterials play an important role in this field, and, in particular, among metallic materials, silver nanoparticles were demonstrated to be effective against a wide range of viruses, in addition to having a strong antibacterial effect. Although the mechanism of antiviral action is not completely clarified, silver nanoparticles can directly act on viruses, and on their first steps of interaction with the host cell, depending on several factors, such as size, shape, functionalization and concentration. This review provides an overview of the antiviral properties of silver nanoparticles, along with their demonstrated mechanisms of action and factors mainly influencing their properties. In addition, the fields of potential application are analyzed, demonstrating the versatility of silver nanoparticles, which can be involved in several devices and applications, including biomedical applications, considering both human and animal health, environmental applications, such as air filtration and water treatment, and for food and textile industry purposes. For each application, the study level of the device is indicated, if it is either a laboratory study or a commercial product.

## 1. Introduction

Emerging and re-emerging viruses represent a growing threat to human health. Viruses are nanometer-sized obligate intracellular pathogens that cause a wide range of adverse conditions in humans, but also in animals and plants, with a tremendous impact on society in terms of morbidity, mortality and economic burden [1,2]. Nonetheless, the list of viral diseases for which antiviral therapies or vaccines are available is still relatively short and some viruses are developing resistance to current therapies [3]. In particular, approved antivirals are mostly directed against chronic infections caused by the immunodeficiency virus (HIV-1), hepatitis B virus (HBV), hepatitis C virus (HCV), herpes simplex viruses type I and II (HSV-1 and HSV-2) and cytomegalovirus (CMV), and a few drugs were approved to treat acute infections, mainly influenza [3]. Vaccination remains the most effective tool in controlling infectious viral diseases, such as hepatitis A and B, rotavirus gastroenteritis, mumps, varicella and the current coronavirus disease (COVID-19), but there are still numerous viruses for which we do not have any preventive or therapeutic tool [3,4]. In this context, and considering the high prevalence of viral infections, research on new antiviral drugs or new preventive antiviral strategies has become extremely important and a target of intense scientific interest.

In the last decade, nanotechnology has shown promise in fighting viruses, and in particular, silver nanoparticles (AgNPs) have attracted the attention of the scientific community due to their wide-spectrum antimicrobial activity and their potential applications in different biomedical fields. AgNPs have a typical range in size between 1 nm and 100 nm. A large variety of AgNP synthesis techniques, such as chemical, physical and biological methods, exist [5] which can significantly influence their size, structure and properties. Nevertheless, numerous factors, such as dispersing agents, surfactants and temperature, can be controlled in order to obtain AgNPs with certain sizes and properties [6].

A large body of literature reports that AgNPs are endowed with antibacterial and antifungal activity against, for example *Staphylococcus aureus*, *Escherichia coli* and *Candida albicans* [7,8,9], and indicates that these properties depend on the ability of AgNPs to directly bind microorganisms, thus having a direct biocidal effect, or an effect on their ability to alter DNA and protein functions. Furthermore, some work has shown that Ag+ cations (that can be released by AgNPs) have a strong affinity for organic components containing sulfur and phosphorus atoms, demonstrating that the antimicrobial effect of silver is due to the interaction of these ions with thiol groups, with a consequential impairment of the dimensional stability of key functional proteins [10,11,12,13]. More recently, numerous studies have also reported the antiviral activity of AgNPs against a wide range of virus families and have demonstrated their particularly interesting preventive action, mainly consisting of a direct virucidal activity or an alteration of the virus binding to the host cell. In 2005, Elechiguerra et al. were the first authors to describe the interaction between HIV and AgNPs and the relationship between the anti-HIV effect and particle size [14]. From that moment, several studies have been conducted with the aim of investigating the antiviral properties of AgNPs. Viruses belonging to different categories, such as enveloped and naked viruses, DNA and RNA viruses, and well-known and emerging viruses, have been screened, with influenza virus type A (IFV) [15,16,17,18,19,20,21] and HSV-1 and HSV-2 [22,23,24,25,26] being the most investigated. Factors influencing the antiviral activity of AgNPs are similar to those recognized for their antibacterial effect, such as size, shape, distribution/concentration and surface chemistry and morphology [27,28,29,30,31].

Based on the existing evidence, different applications of AgNPs in the virology field have been proposed, such as viral disease treatment and prevention, veterinary applications, water and air treatment, and food and textile industry applications. To the best of our knowledge, a complete review describing the most recent findings on the antiviral activity and potential application of AgNPs is missing. Therefore, this review aims to analyze the wide-spectrum antiviral properties of AgNPs, along with their mechanisms of action, the factors influencing their antiviral effects and the validated or up-and-coming applications. We should mention that there are also many promising antiviral agents based on gold nanoparticles, silicon nanoparticles, zinc oxide, copper nanoparticles and iron oxide nanoparticles—just to mention a few—that are the subject of other interesting peer-reviewed works [32,33,34].

## 2. Wide-Spectrum Antiviral Activity of AgNPs

Table 1 summarizes the most relevant studies published since the year 2010 relative to the antiviral activity of AgNPs. In the first column of Table 1, we report the viruses (and their related viral families) against which AgNPs have been tested, either in vitro and/or in vivo, as specified in the second column. Moreover, we report the type of AgNPs, a simple expression giving a short description of the antiviral mechanism of action, as well as the factors influencing the antiviral effect: the expression “↑C.→↑A.e.” means that by increasing AgNPs concentrations, the antiviral effect increases; “↑E.t.→↑A.e” means that by increasing the exposure time, the antiviral effect increases.

### 2.1. In Vitro and In Vivo Evidence of AgNP Antiviral Efficacy

The results of our electronic search highlighted the wide-spectrum antiviral potential of AgNPs. Indeed, it was reported that the infectivity of 31 viruses, belonging to 17 different virus families, was reduced by AgNPs. Notably, the most recent findings indicated that AgNPs are also endowed with antiviral activity against SARS-CoV-2 [36] and suggested that they could find application as a complementary strategy to ending to the current pandemic [63]. Other airborne viruses that, similar to SARS-CoV-2, can spread efficiently among humans, causing outbreaks that are difficult to control [62], have been investigated. In particular, numerous in vitro studies reported the anti-IFV activity of AgNPs of different origins [15,16,17,18,20,21,35], and some reports also indicated their activity also against the respiratory syncytial virus (RSV) [45,46], human parainfluenza virus (HPIV) [22] and adenovirus (ADV) [59]. Interestingly, the anti-IFV and anti-RSV activity of AgNPs was also demonstrated in in vivo models. Dongxi et al. demonstrated that intranasal AgNPs administration in mice significantly enhanced survival after infection with the H3N2 IFV, showing lower lung viral titers and minor pathologic lesions in lung tissue [20]. Morris et al. studied the effect of AgNPs on RSV experimentally infected BALB/c mice. They reported an AgNP-mediated reduction of RSV replication in mice, with significant reduction in pro-inflammatory cytokine (i.e., IL-1α, IL-6, TNF-α) and pro-inflammatory chemokine (i.e., CCL2, CCL3, CCL5) levels. They also observed an increase of neutrophil recruitment and activation in the lung tissue [45]. To the best of our knowledge, no reports showed the antiviral potential of these nanoparticles against the widely diffused rhinoviruses (HRVs), the leading cause of the common cold. We recently demonstrated consistently that a silver nanoclusters/silica composite coating deposited via co-sputtering technique on air filters as well as on cotton textiles was significantly active against RSV and IFV, but not against HRV [63]. This evidence highlights that the antiviral effect of AgNPs on this latter respiratory pathogen is still an open question. 

Along with respiratory viruses, HSV-1 and HSV-2 were deeply investigated for their sensibility to AgNPs. The anti-HSV activity of functionalized AgNPs [24,25] and plant- or fungi-derived AgNPs has been analyzed in vitro [22,23], and in vivo results were produced too. In particular, tannic acid-modified AgNPs (TA-AgNPs) were tested in mouse models of vaginal HSV-2 infections [24], and the results indicated that they induced production of cytokines and chemokines important for an antiviral response, improved the clinical outcome of the mice and reduced virus titers in the vaginal tissues. Following a re-challenge, the vaginal tissues treated with TA-AgNPs showed a significant increase in the percentages of IFN-gamma+ CD8+ T-cells, activated B cells and plasma cells, while the spleens contained significantly higher percentages of IFN-gamma+ NK cells and effector-memory CD8+ T cells in comparison to the NaCl-treated group. Szymanska et al. [25] developed a mucoadhesive gelling system with TA-AgNPs for the potential treatment of HSV-2 infections, and they consistently confirmed the efficacy in in vivo models. Despite this evidence, further studies are undoubtedly needed to clarify the role of nanoparticles in preventing HSV infections in vivo. AgNPs have also proved active against sexually and blood-transmitted viruses, such as HIV, HBV and HCV [43,50,60]; against viruses transmitted via the fecal–oral route, such as the hepatitis A virus (HAV) and enteroviruses [23]; and against emerging arboviruses, such as the Chikungunya virus (CHIKV) and Dengue virus (DENV) [48,49,53]. It should be outlined that only a few in vitro reports are available for each virus, and that the efficacy of AgNPs against the latter-mentioned viruses needs further investigation.

### 2.2. Antiviral Mechanisms of Action of AgNPs

Generally, antiviral compounds inhibit the formation of new viruses by blocking crucial steps of viral replication or directly acting on the virus [3]. Understanding of the antiviral mechanism of action of AgNPs is therefore essential for target-oriented and efficient development of new antiviral strategies exploiting nanoparticles. Nevertheless, their antiviral mechanisms are still under investigation and seem to depend on the synthesis of AgNPs, with many factors such as size, shape and surface functionalization influencing the antiviral activity, as discussed in Section 2.3. Part of this review is addressed to clarify the principal inhibitive mechanisms that have been outlined so far. Numerous studies have demonstrated that AgNPs mainly act by physically interacting with the free viral particle or with the cell-bound viral particle. In this way, AgNPs can exert a virucidal activity—i.e., an inactivation of the infectious viral particle or a morphological alteration of the virion; or can inhibit the early phases of viral replication—i.e., the virus binding to the host cell or virus penetration processes. Although these are the most represented antiviral mechanisms, inhibition of the late steps of viral replication has also been proposed, but these effects at the intracellular level are less known. From this perspective, the broad-spectrum antiviral activity of AgNPs, along with their ability to prevent cell infection, has attracted strong attention in many sectors, starting from water and air treatment, personal protective equipment production, food packaging, the textile industry to the biomedical field, which is probably one of the fastest-growing areas. Indeed, the disruption of viral particles or the inhibition of viral attachment and entry processes are very attractive strategies from a medical point of view. The possibility of preventing cell infection would first limit the toxicity for the host and the risk of inducing viral resistance, and would be ideal in order to offer an efficient and immediate strategy in the fight against new, emerging viral strains. 

The mechanisms of action of the AgNPs studied so far are reported in the fifth column of Table 1 (when available), and they will be described in the next subchapters in detail. Table 2 reports the main mechanisms of action of AgNPs related to the affected viruses. Nonetheless, it should be noted that there is a heterogeneity in the experiments employed to clarify AgNPs’ mechanisms of action, sometimes making it difficult to define an inhibited step of viral replication. Numerous studies clearly demonstrated the antiviral activity of AgNPs and sometimes showed their physical interaction with a virus, but they only suggested a possible mechanism of action and indicated that their results require further investigations.

#### 2.2.1. Virucidal Activity of AgNPs

The term “virucidal activity” refers to the ability of an antiviral compound to permanently inactivate or morphologically alter the infectious viral particle. Both spherical and functionalized AgNPs have been investigated. Spherical AgNPs have been demonstrated to exert a virucidal activity against IFV-A, ADV and the Bunyamwera virus (BUNV) [20,59,61]. Free infectious viral particles were pre-incubated with AgNPs for different periods and the viral morphological structure was subsequently observed by the transmission electron microscope (TEM) and compared to the untreated control. The results clearly showed that AgNPs can directly interact with the virus and can destroy the viral morphological structure, leading to the inhibition of viral function in a time-dependent manner. Oseltamivir-modified AgNPs also showed a similar mechanism of action against IFV-A, inducing morphologic abnormalities in the viral particle and inactivating it [16]. Consistently, Yang et al. demonstrated that the pre-treatment of RSV with curcumin-modified AgNP (cAgNP) determined a significant reduction of the viral titer compared to the other type of treatment (cell-pretreatment or post-infection treatment). Furthermore, through dynamic light scattering (DLS) measurements, the authors showed an increase in diameter in cAgNP-treated RSV, suggesting a direct attachment of AgNPs to the viral envelope glycoproteins [46]. 

A few reports indicated that AgNPs can inactivate the viral particle, damaging crucial viral components such as glycoproteins on the viral envelope or structural proteins of the viral capsid. AgNP-decorated silica hybrid composites (Ag30–SiO2) effectively inhibited IFV-A and were shown to interact with the two viral glycoproteins located at the virus envelope, hemagglutinin (HA) and neuroamidase (NA), significantly affecting their activity after 1 h of virus exposure. The authors concluded that Ag30–SiO2 composites were able to reduce IFV-A infection in MDCK cells due to their interactions with the outer membrane of this virus [17]. Additionally, Bekele et al. demonstrated that the antiviral activity of AgNPs against feline calicivirus (FCV) was due to direct damage of the viral capsid protein. The protein integrity in viral suspension treated with AgNPs was evaluated through WB analysis: AgNP-treated suspension of FCV particles showed a reduction in the Western blot signal of the viral capsid protein compared with the control, indicating that nanoparticles can directly affect the virion structure [27].

Preliminary and recent results on SARS-CoV-2 [36] suggested that AgNPs were able to inhibit extracellular free virions in a size-dependent manner, but the precise mechanism of action against this virus remains to be elucidated. 

The virucidal activity of AgNPs is an interesting characteristic that could be widely exploited for the production of different devices to limit the spread of a broad spectrum of viruses. At the end of this review, we will analyze the ideal characteristics that an antiviral material should have in order to exhibit a virucidal effect without causing problems for human health or for environmental safety.

#### 2.2.2. AgNP Inhibition of the Early Steps of Viral Replication 

AgNPs have also been shown to inhibit the early steps of viral replication, i.e., the virus binding to the host cell or the virus cell-penetration processes (entry step). The inhibition of viral binding refers to the ability of an antiviral compound to transiently interact with viral receptors, thereby impeding the interaction with cells, or to compete with the virus for the same cellular receptors, thus masking them and avoiding virus attachment to cells. On the contrary, the inhibition of the viral entry step consists in the specific alteration of one of the different entry processes that viruses can exploit to infect cells, including viral envelope fusion with the cell membrane or receptor mediated-endocytosis. The use of different temperatures—−4 °C and 37 °C—allows for an in vitro examination of the effect of AgNPs on each specific step of viral infection.

Tannic acid-modified AgNPs (TA-AgNPs) have been extensively investigated against HSV-2. Orlowski et al. demonstrated that TA-AgNPs were capable of reducing HSV-2 infectivity both in vitro and in vivo [24,44]. In particular, they clearly demonstrated that TA-AgNPs of different size prevented the attachment of HSV-2 by 40–80%, with the highest tested size (46 nm) being the most effective. The same nanoparticles were also able to impair virus entry with approx. 80% efficiency, and with no size-related differences. In addition, through morphological and compositional analyses, the authors showed that AgNPs interact with the virion’s surface, creating a physical obstacle that impairs the interaction of viral receptors with the cell surface. Mucoadhesive gelling systems with TA-AgNPs were then developed to create an effective topic system to treat HSV infections [25]. The results from the study of their mechanism of action confirmed the ability of different nanoparticle-doped hydrogels to affect viral attachment and impede penetration, although profound differences in the activity evoked by the tested preparations toward HSV-1 and HSV-2 were noted. HSV-1 was substantially impaired in its attachment to cells, while HSV-2, against which TA-AgNPs showed more potent activity, was significantly impaired in its entry inside the host cells. Additionally, another type of functionalized AgNP was tested in vitro against HSV-1: AgNPs capped with mercaptoethane sulfonate (MES) were designed with a view to inhibiting the early replication phases of the virus [26]. Through their sulfonate end groups, AgNPs with MES were predicted to compete with the virus in its binding to cellular heparan sulfates (HS), key receptors for HSV-1 binding to cells. The authors demonstrated an effective inhibition of HSV-1 infection in cell cultures by the capped nanoparticles, but their precise mechanism of action remains to be elucidated. Interestingly, since numerous viruses are HS-dependent for their attachment to cells, the use of this particular functionalization for the prevention of cell infection could be extended to other pathogens.

The inhibition of viral entry was also demonstrated in two reports investigating the antiviral activity of both chemical and plant-synthetized AgNPs. Uncoated AgNPs were capable of inhibiting vaccinia virus (VACV) entry in a dose-dependent fashion: the virus was indeed able to adsorb to cell receptors, but it could not enter cells in the presence of AgNPs. Additionally, when macropinocytosis was inhibited (via Pak1 silencing), the antiviral effect of AgNPs was significantly reduced, suggesting that macropinocytosis was required for the full antiviral effect [57]. On the other hand, AgNPs biologically synthetized from silver nitrate using an aqueous extract of the Argimone maxicana significantly impaired the infectivity of peste des pestits ruminants virus (PPRV) at the level of virus entry [47]. These AgNPs were demonstrated to interact with the virion surface as well as the virion core without having a direct virucidal activity but exerting a blocking effect on viral entry into the target cells. The inhibition of virus binding and entry can be considered an effective preventive strategy. Nevertheless, since nanoparticles and viruses have to be simultaneously present at the host cell surface, the potential toxic effect of AgNPs must be analyzed in detail before developing a potential application.

#### 2.2.3. AgNP Inhibition of the Late Steps of Viral Replication

A few reports indicated the ability of AgNPs to act on the late steps of viral replication of two different hepatitis viruses. Recently, Shady et al. reported that green-synthesized AgNPs from total extract and petroleum ether fraction of Amphimedon sp exerted an inhibitive action on HCV NS3 helicase and protease, a key enzyme for HCV replication, and that this activity was superior to the silver nitrate AgNPs used as the control [50]. These data were obtained through an anti–HCV helicase and protease biochemical assay; therefore, more in-depth studies, including cell-based assays and in vivo assays, are required to confirm the result. Conversely, Lu et al. attributed the anti-HBV activity of AgNPs to their ability to directly interact with HBV double-stranded DNA or the viral particle. The authors showed that AgNPs (sized 10 and 50 nm) were able to reduce the extracellular HBV DNA formation of HepAD38 cells by >50% compared with the vehicle control and that they could inhibit the formation of HBV RNA. Gel mobility shift assays indicated that AgNPs bound HBV double-stranded DNA at a 1:50 molar ratio of DNA: silver and that binding affinity for HBV DNA was confirmed through an absorption titration assay. Additionally, TEM analysis revealed that AgNPs could also directly interact with HBV viral particles obtained from patients [60].

Generally, we can conclude that the inhibition of the late steps of viral replication was rarely observed and is certainly not the main antiviral mechanism of action of AgNPs.

### 2.3. AgNP Factors Influencing Virucidal Activity 

AgNPs can be synthesized via several methods, including physical, chemical and biological processes, obtaining particles with different structure, morphology and surface chemistry. A complete overview of the several synthesis methods is outside of this review’s topic, but they are well described in [64], in which Salem et al. provided an updated digression concerning the different methods used to obtain AgNPs with different properties. According to this review, we have identified some specific factors and features that influence the antiviral properties of AgNPs, including size, concentration, shape, surface chemistry and functionalization and exposure time, as reported in the sixth column of Table 1.

#### 2.3.1. Impact of Size and Concentration on AgNP Antiviral Activity 

AgNP antiviral activity is predominantly size- and dose-dependent, as indicated in Figure 1.

AgNPs are usually synthetized with a dimension of less than 100 nm. However, only the size range between 5 and 30 nm is considered acceptable for exploiting an antiviral effect, whose maximum is usually achieved with AgNPs of less than or equal to 10 nm. Smaller particles act more effectively towards viral infection thanks to the high mobility and the size rather than the virus dimension, as reported in the studies on H1N1 influenza A virus [19,20,21], poliovirus [39], HIV [14], HSV-1 and HSV-2 [26]. Elechiguerra et al. demonstrated that AgNPs in a size range of 1–10 nm interacted with HIV-1, while larger particles did not bind the virus [14]. The binding of HIV to cells occurs through glycoproteins arranged with a regular spatial distance on the viral envelope, such that only nanoparticles with a certain size comparable to this distance can bind virus proteins and inhibit their replication. This study suggests that a similar mechanism of action exploiting particle size could be applied to other viruses.

Huy et al. indicated that the difference in size between AgNPs and viral particles is also important to determine the antiviral effect. In fact, considering different concentrations of AgNPs and different amounts of poliovirus infectious particles, AgNPs with a size of 7.1 nm reduced viral infectivity events at the lowest tested concentration. This was probably influenced by the nanoparticle’s dimension, which is much smaller than the poliovirus, which is 25–30 nm. This difference in size conferred a large mobility on silver nanoparticles, and thus an easy interaction with poliovirus particles [39]. In an in vitro study, similar AgNPs of about 9 nm, obtained via electrochemical method, impaired IFV infection, interacting with viral particles and destroying their morphological structures in a time-dependent manner [20]. The same AgNPs, tested in vivo via an intranasal administration, inhibited virus replication in the lungs and the development of pathologic lung lesions [20]. Jeremiah et al. evaluated the antiviral potential of commercial AgNPs, with a dimension range between 2 and 100 nm, against SARS-CoV-2 and confirmed the strongest effect of the smaller nanoparticles [36]. In a similar way, Bekele et al. compared AgNPs of different size (10, 75 and 110 nm), finding that only 10 nm particles showed antiviral activity towards FCV, while 75 nm and 110 nm nanoparticles did not reduce the viral titer [27]. 

Particles of less than 10 nm and with a more controlled dimension range (less standard deviation) are usually produced via electrochemical methods, whereas biological processes generally produce nanoparticles with a larger size range. Despite this, the antiviral activity of the green-synthesized AgNPs was demonstrated. The AgNPs obtained from *F. oxysporum* (4–13 nm) and Curvularia (5–23 nm) species are more effective against HSV-1 and HPIV-3 viruses than AgNPs obtained by Alternaria (7–20 nm), Phoma (7–20 nm) and *C. indicum* (10–31 nm), which have a bigger average size than the previous types [22]. Murugan et al. synthesized AgNPs using extracts from an alga, *C. clavulatum*, or the leaves of a small tree, *B. cylindrica*, obtaining spherical nanoparticles with a dimension range between 35–65 nm and 30–70 nm, and successfully tested them against DENV [51,52]. Similar results were obtained using particles with an average size of 100 nm, obtained using Moringa oleifera extract as a reducing and stabilizing agent [53]. The bigger dimensions of nanoparticles obtained by green synthesis methods were confirmed by the work of Sharma et al. [48,49], who obtained AgNPs from Psidium guajava leaf extract, with a size of about 100 nm, and synthesized AgNPs with an average size between 50 and 120 nm using medicinal plants *A. paniculata*, *P. niruri* and *T. cordifolia*. In both cases, the nanoparticles obtained were stable and effective against CHIKV, despite the size being greater than 10 nm.

Size plays a synergic role together with AgNP concentration in modulating antiviral activity [24]. Larger AgNPs must be used in low concentrations to avoid cytotoxicity [23,48,49,51,52]. It was possible to obtain similar effects against HSV-2 using AgNPs with a size of 33 nm and concentration of 0.5–1 µg/mL, AgNPs of 13 nm with a concentration of up to 1 µg/mL and larger nanoparticles of 46 nm with a higher concentration of 2.5 µg/mL. Nevertheless, in this latter case, there was an increased risk of cytotoxicity [24]. Based on this premise, the right balance between size and concentration can also produce antiviral activity with larger nanoparticles. In general, the higher is AgNPs concentration, the greater is the antiviral effect. Most studies reported the antiviral efficacy of AgNPs at concentrations ranging between 10 and 100 ppm [65] but smaller AgNPs and higher concentration increase the risk of toxicity towards cells [39].

#### 2.3.2. Impact of AgNP Functionalization on Antiviral Activity

Several approaches, such as AgNP surface modification, coating of only AgNPs or combinations with other elements, could be used to prevent direct contact between metallic nanoparticles and cells, reducing in this way their possible cytotoxicity. However, these methods can also influence the interaction between viruses and nanoparticles: the antiviral effect of functionalized or coated AgNPs or AgNPs embedded in a composite material could be affected by physical impediments due to the combination with a second element. As an example, metallic nanoparticles, encapsulated in organic or non-organic materials such as chitosan, collagen or gelatin, demonstrated good antiviral activity with a prolonged effect thanks to the slow ion release [66]. Mori et al. developed an environmentally friendly composite system of AgNPs in a chitosan matrix, demonstrating that the antiviral effect on IFVA depended only on the presence of AgNPs [21]. The amount and the size of the AgNPs influenced the antiviral activity of the composite system, which increased with a higher concentration and smaller dimensions of AgNPs, whereas the chitosan matrix did not show any antiviral effects. On one side, the chitosan matrix could weaken or hinder the direct contact between the virus and the nanoparticles. On the other side, since the matrix constitutes a further physical impairment, virions are bonded to the composite material and the interaction with the host cells is prevented.

Elechiguerra et al. [14] studied the effect of surface functionalization of AgNPs against HIV. Besides considering the influence of AgNP size, as previously mentioned, the second goal of this work was to evaluate the anti-HIV effect of AgNPs, comparing three different approaches: AgNPs encapsulated in a foamy carbon matrix as composite material, AgNPs coated in poly (N-vinyl-2-pyrrolidone) (PVP) and AgNPs modified with bovine serum albumin (BSA) as surface functionalization. BSA-functionalized and PVP-coated nanoparticles showed lower inhibition efficacy compared with nanoparticles embedded into the carbon matrix. In fact, the first two systems modified the AgNP surfaces in terms of chemical and morphological properties, respectively. The nanoparticles were in direct contact with the capping agent and not with the virus, while the nanoparticles released from the carbon matrix due to their weak interaction exposed free surface area and came into direct contact with the virus. This confirmed that the presence of a coating or a surface functionalization on AgNPs can influence interactions with viruses. Another system, constituted by an aqueous colloidal solution of AgNPs capped with the mercaptoethane sulfonate group (Ag–MES), was tested to prevent replication of HSV-1 [26]. Considering the low virus concentration, infected cells, treated with the Ag–MES system, did not show any morphological change, whereas infected and untreated cells showed an increase in HSV-1 plaque diameter. On the other hand, higher virus concentration causes small plaques on cells treated by Ag–MES, and the complete destruction of the non-treated cell monolayer. The same results were found for HSV-2 [67]. This suggested that the Ag–MES system acted as an inhibitor as it emulated cellular receptors, such as heparan sulfate (HS), inducing the virus to bind AgNPs. These nanoparticles have been applied to an HSV-1 inhibition test, but they potentially can be used to hinder other viruses, since functional groups present on the nanoparticle surface and different basic materials can attract and inhibit different types of viruses [26].

We could conclude that it is possible to increase the antiviral activity of silver nanoparticles depending on the element or system used for their functionalization. If the modification approach has an intrinsic antiviral effect, as in the case of tannic acid [24,44], the synergetic effect of AgNPs and the modifier agent can directly increase the antiviral activity of the system. In other cases, the presence of a second element or a matrix, although not intrinsically antiviral, could confer peculiar characteristics on the system that result in increasing its antiviral effect. Considering the case of Ag–MES nanoparticles, MES alone was ineffective in inhibiting HSV-1, but when employed as functionalization, it showed an ability to mimic the cellular HS groups, promoting interaction between virus and modified AgNPs and inhibiting virus replication [26]. The same consideration concerns the chitosan matrix/AgNPs system: chitosan is not antiviral itself, but if used as a matrix for nanoparticles, it represents a physical impairment for viruses, consequently amplifying the effect of the nanoparticles [21]. Regarding the choice of the system to be used, it is necessary to consider and to test the effect of the precursor molecules and byproducts on human or animal safety, or on the environment, depending on the application.

#### 2.3.3. Impact of AgNP Shape on Antiviral Activity 

Another feature that influences the antiviral activity of AgNPs is the nanomaterial shape. In general, in the virology field, silver nanomaterials are produced with a spherical or quasi-spherical shape, but it is possible to obtain nanomaterials such as nanowires or nanorods. In this case, in addition to the effect of the size, it is interesting to evaluate the dependence of the antiviral effect on the shape. This was investigated by Xiaonan et al., who compared AgNPs with a size of less than 20 nm and Ag nanowires with a diameter of 60 nm and 400 nm, demonstrating a significant antiviral activity towards the transmissible gastroenteritis virus (TGV), in contrast with Ag colloids, which did not show any effect [30]. This can be explained by considering that silver colloids were necessarily dispersed into polyvinylpyrrolidone (PVP) for the synthesis and dispersion and in direct binding with it, avoiding in this way the direct contact between silver and cells or viruses. In addition, considering the same concentrations of AgNPs and nanowires, the best reduction in virus titer is achieved with the smallest particles, emphasizing the importance of nanoparticle shape and dimension. 

The available studies were mainly conducted on spherical nanoparticles, while the effect of different shapes, such as nanowires, was not deeply investigated. It probably depends on the evidence that spherical nanoparticles have high surface area-to-volume ratios, which play a crucial role in the antiviral effect, since they make the interaction between nanoparticles and viruses easier. However, it could be interesting to investigate in detail if and how other nanomaterial shapes can interact with viruses and inhibit their replication, in order to have an alternative to silver nanoparticles for antiviral applications.

## 3. Application Fields of AgNPs

Silver nanoparticles are considered interesting materials for their great versatility, which allows involving them in different types of applications, from the biomedical field [68] to the electronics industry [69] and agriculture field [70]. Figure 2 represents a summary of the different synthesis methods of AgNPs, their effects and possible fields of application. 

As mentioned before, the focus of this review will be AgNPs’ antiviral effects and their application in this field.

Since the antibacterial properties of silver have been known for many years, many commercial products have been developed to exploit this peculiarity, from home textiles (Microban^®^ [71] SilverShield^®^ [72]) to underwear (Silverskin^®^ [73]) and antibacterial paints (Icosan Defend AG [74]). On the contrary, the most recent discovery about the antiviral activity of silver has not yet allowed the development of numerous products. Only with the recent COVID-19 pandemic have some products, such as masks or sprays for surfaces using AgNPs and other elements or technology, been put on the market to counteract the SARS-CoV-2 epidemic; for example, the FFP2 face mask (Nanosilver^®^ [75]). However, antiviral AgNPs could be used in the development of products and devices in other fields with a high risk of viral infection. In this review, we individuated and described the main potential applications of antiviral AgNPs in health and medical applications in terms of potential therapeutic use, veterinary use, applications in water treatment and air filtration systems, food packaging and textile industry purposes (Figure 3).

Table 3 summarizes the fields of application of antiviral AgNPs and a description of the AgNP system, indicating if it is an embryonic study at lab scale (indicated as laboratory study) or could be considered a prototype or a commercial product.

### 3.1. Application in the Human Health Sector

Considering the human health applications, AgNPs could be used for the treatment and prevention of several viral diseases. Nevertheless, despite their antiviral activity being demonstrated against numerous viruses, the majority of studies refer to in vitro results (as outlined in Table 1) and they are still far from the development of an effective therapeutic drug for humans. In Table 3, we reported, under the term “Potential therapeutic use”, only the studies that reached the first step of in vivo analysis [20,24,25,44,45,56]. The lack of evidence in vivo and the absence of clinical trials outline that the therapeutic/prophylactic field is not the main potential application of AgNPs at present. The principal obstacle is probably their possible toxicity in vivo, i.e., their potential accumulation in different body sites with long-term sequelae. Further investigations are needed to analyze these aspects and to open the way to the development of new antiviral drugs.

On the other hand, the prevention of viral infection is an important feature to be considered and the application of AgNPs in this field has reached interesting and developable results. 

A commercial product called Argovit^TM^ demonstrated a dose-dependent reduction of RVFV infectivity [56]. The same system was recently tested against SARS-CoV-2 in humans [37]. Almanza-Reyes et al. tested the effect of mouthwash and nose rinse using Argovit^TM^ on a group of 231 participants, including men and women, in a range of age between 18 and 65 years old and with different occupations, divided randomly into two groups. The first group, named the experimental group, used a spray containing an AgNP solution mixed with water to gargle three times a day and do nasal lavages, while the second group, named the control group, used conventional mouthwash for a nose rinse and mouthwash. The results showed that in the experimental group, the number of SARS-CoV-2 infections was lower than in the control group. Therefore, the authors emphasized the importance of oral and nasal hygiene to a reduction of the risk of SARS-CoV-2 infection in health personnel who are exposed to patients diagnosed with COVID-19. 

Additionally, since some viruses can be transmitted indirectly through contaminated surfaces, the development of self-decontaminating or antiviral surfaces is needed. Jan Hodek et al. tested in vitro protective hybrid AgNP coatings prepared by the sol-gel method against HIV-1, DENV, HSV-1, IFVA and COXB3 viruses (enveloped, non-enveloped, RNA and DNA viruses) [76]. The coatings, deposited on substrates such as glass and Poly(methyl methacrylate) (PMMA), reduced the infectivity of all of the above-mentioned viruses, with the exception of COXB3. This selective effect could be ascribable to the different composition of virus capsids and outer membranes and the consequential mechanisms of interaction between AgNPs and viruses. The authors showed that the system was endowed with virucidal activity against different enveloped viruses, thus suggesting that this hybrid coating, which was previously demonstrated to be active against some bacteria strains, has the potential to provide antimicrobial protection on surfaces and materials in healthcare settings. Another preliminary study focused attention on the application of AgNP-coated condoms for the prevention of sexually transmitted diseases [77]. Common condoms generally constitute a physical barrier to sexually transmitted diseases, but in this case, they have an intrinsic ability to inactivate some viruses, bacteria and fungi. Since AgNPs showed marked antiviral effect against HIV, HSV-1 and HSV-2 [14,22,24,25,29,67], Fayaz et al. developed AgNP-coated polyurethane condoms by immersion in an aqueous silver solution [77]. AgNP-coated condoms, with stable nanoparticles (not removed by water washes), were directly exposed to these viruses, showing a significant and time-dependent reduction of viral infectivity. AgNP-coated polyurethane condoms can directly inactivate viral particles, and could improve the protection given by common condoms as an important first-line product against sexually transmitted infections. 

In the context of viral disease prevention and protection, the discussion can also be extended to the personal protective system (PPE). Recently, the spread of SARS-CoV-2, which generated the COVID-19 pandemic, increased the use of individual protective systems, as recommended by the World Health Organization (WHO). Traditional PPE systems have not had an intrinsic antiviral activity but have guaranteed good temporary protection. Many AgNP-doped or -coated masks have been commercialized with a specific antiviral effect (Nanosilver^®^ [75]). A composite coating, consisting of silver nanoclusters in a silica matrix, was deposited via co-sputtering through a patented process on a face mask of type FFP3 [78]. The coating was able to effectively reduce, in vitro, the titer of the SARS-CoV-2 virus to zero after 1 h 30 min of incubation. Surgical masks were also doped with AgNPs by means of immersion in an aqueous solution containing nanoparticles obtained via an electrochemical method [32]. Considering the crucial role of face masks during the pandemic period, an important aspect that must be considered is the modification of their filtration performances. Indeed, different quality factors were evaluated, showing that the presence of an antiviral coating on an air filter system did not affect the filtration quality [63]. This solution demonstrated the effective inactivation of IFVA, another potential pandemic virus, suggesting a possible future application for respiratory devices and clinical textile materials. 

AgNP-doped graphene oxide (GO) sheets were developed for personal protective equipment to decrease virus transmission, in particular for respirators and masks [37]. Besides its well-documented antibacterial properties [91] graphene oxide was chosen for its peculiar characteristics of high carrier mobility, large surface area and biocompatibility, which prevent agglomeration but help in the grafting of AgNPs on the graphene sheets, limiting the risk of toxicity. Both GO and AgNP-doped GO sheets, with an average AgNP size of 7.5 nm, were tested towards an enveloped virus, feline coronavirus (FCoV), which belongs to the Coronaviridae family, and a non-enveloped virus, the infectious bursal disease virus (IBDV), which belongs to Birinaviridae family. The researchers selected these two animal viruses only for their strict regulation in terms of handling as they do not provide zoonotic transmission. Both GO and AgNP-doped GO sheets inhibited the infectivity of FCoV after one hour of incubation, whereas only AgNP-doped GO sheets demonstrated antiviral efficacy towards IBDV. Despite the mechanism of action remaining to be elucidated, the authors hypothesized that the antiviral effect of GO sheets against enveloped viruses could be due to a strong physicochemical interaction between GO and the lipids of the viral envelope [28]. Regarding naked viruses, due to the absence of the lipid membrane, the same mechanism cannot occur. In this case, the antiviral activity was attributed to the AgNPs, with GO sheets improving particle distribution, preventing the formation of agglomerates and increasing antiviral Ag properties. This type of coating could be applied to face masks in order to improve the efficacy of N95 and three-layer surgical masks, whose antimicrobial effect is reduced in the presence of water or moisture.

Currently, the use of AgNPs for viral disease preventive strategies could be limited by the absence of methods to recollect the nanoparticles or to recycle and re-use the product in order to reduce waste and avoid the use of chemicals involved in the production of new systems.

### 3.2. Veterinary Applications of AgNPs

Another potential field of AgNP application is animal health, since many viruses affect animals, causing high morbidity and mortality [92]. This has repercussions for the economy and for human life, because many sources of livelihood, such as eggs, milk, meat, fiber and other foods, are animal derivatives. To the best of our knowledge, there are only a few studies investigating the potential use of AgNPs as antiviral agents in animals. However, we would like to emphasize that these studies, as well as studies on antiviral AgNPs for humans, were performed in vitro, with application only hypothesized and still far from reality. As an example, AgNPs were tested against the peste des petits ruminants virus (PPRV), a virus highly contagious in small ruminants [47]. The vaccine against this virus turns out to be ineffective and insufficient to prevent contagion between animals and to avoid the serious economic losses that it causes. Khandelwal et al. demonstrated that AgNPs act by altering the entry of PPRV into the host cells in vitro at nontoxic concentration, and they suggested that the use of AgNPs can be explored as a new antiviral therapy against this virus.

The infectious bursal disease (IBDV) virus is another example of a virus affecting animals, generally on poultry farms, and one that can be transmitted through direct contact or through water or contaminated fertilizers. Pangestika et al. [62] demonstrated, in preliminary experiments, that AgNPs are also endowed with antiviral activity against this virus at concentrations of 20 ppm. Recently, attention was focused on a virus that mainly affects dogs: the canine distemper virus (CDV). Although vaccines are available, dogs are not protected for their entire life. Bogdanchikova et al. [79] tested a treatment with a solution of AgNPs (6%) and PVP (94%), in quantities depending on the dog’s weight, on dogs affected by CDV with a neurological and non-neurological infection. Other dogs were treated with traditional methods and were considered as controls. The author showed that treatment with AgNPs produced good results in the case of non-neurological infections and was not harmful to the dog, while all the control dogs died. For these reasons and considering the low cost and easy handling, the authors suggested that an AgNP and PVP solution can be used as an alternative therapeutic method against CDV.

In the last few years, the aquaculture sector has been characterized by a crisis due to the increasing number of infections by the white spot syndrome virus (WSSV), which is highly infectious and contagious for shrimps. As reported in recent reviews [80,81], several groups of researchers tested AgNPs, including the Argovit system, against WSSV and other pathogens infecting shrimps through intramuscular injection or oral administration, finding AgNPs a promising system for treatment and prevention of diseases affecting marine farms, with consequent losses in the economic field. 

Recently, the World Organization for Animal Health has inserted Newcastle viral disease (NVD) in the category of most significant diseases, which can seriously affect several household and wild bird species, with consequences on poultry production. Two different groups of researchers focused on this topic, with the aim of obtaining silver nanoparticles from natural plant and alga, which are able to inhibit the NVD virus. Mehmood et al. tested in vitro and in ovo the effect of AgNPs obtained from Syzygium aromaticum [82]. The AgNPs obtained could be used as an alternative method for viral infections, in a concentration that allows the obtaining of an antiviral activity without a cytotoxicity effect. 

### 3.3. Applications of AgNPs in Water and Air Filtration Systems

#### 3.3.1. Water Treatment

Water treatment in terms of purification and filtration is fundamental to making clear and potable water. The presence of viruses, such as the adenovirus, rotavirus, norovirus and hepatitis A, was documented both in surface waters and in underground sources [93,94,95,96]. Traditional procedures for water disinfection are chlorine, which led to the production of harmful disinfection byproducts in addition to bad odor or taste, and UV methods, which are not effective on some types of viruses, such as the adenovirus [97], even if they do not form byproducts [98]. An alternative approach can be represented by the use of AgNPs combined with other specific materials, creating systems with increased antiviral effect, without changing the physical and chemical properties of water and limiting the release of NPs into the surrounding environment. As an example, a study demonstrated that AgNPs, produced via L. fermentum, forming the so-called Biogenic Ag0, are able to inactivate murine norovirus-1, and suggested that the inactivation of human noroviruses could also be possible [83]. Another example is titanium dioxide (TiO_2_), which also has the peculiar property of being an environmentally friendly photocatalyst for water treatment. The effect of AgNPs, deposed via photochemical reduction of silver nitrate on TiO_2_, was verified against the bacteriophage MS2 under UV radiation [84]. The bacteriophage MS2 was used by the authors as a model due to its similarities with other waterborne viruses in terms of resistance to chlorine and UV, which is comparable to or higher than those of the hepatitis A virus [99] and poliovirus [100]. Only TiO_2_ inactivated the viruses in 2 min, but the addition of AgNPs to TiO_2_ determined an additional antiviral effect and faster inactivation kinetics. The mechanism can be explained by considering that silver promotes charge separation, causing an increase in the production of ROS, and that virus binds to newly generated HO radicals. Consequently, the adsorption of MS2 on titanium dioxide surfaces was improved, increasing the inactivation rate. The authors propose the use of nAg/TiO_2_ materials for the development of photoreactor or photocatalytic systems for the disinfection of water. An alternative hybrid system was developed using glass-fiber substrates coated by Fe_2_O_3_ and AgNPs (FG-Fe_2_O_3_/Ag) uniformly distributed. This system demonstrated an excellent antiviral activity against the same bacteriophage, MS2, and it could be involved in existing implants to improve disinfection effects [85]. 

Another material widely used in water filtration is activated carbon, thanks to its peculiar nanostructure, high porosity and large specific surface, which make it a valuable absorbent material. Activated carbon filters are not able to neutralize waterborne viruses themselves, but a Ag/CuNP-doped activated carbon filter was tested towards a bacteriophage (T4) selected as a model virus structurally comparable to enteric human viruses [55]. Considering the excellent results in the antiviral field, this filter could be considered a promising material in water purification, thanks to the synergistic performance of activated carbon and Ag or Cu NPs, which act in the interaction between the carboxyl groups of the amino acids present on the virus and the silver, causing the formation of insoluble compounds. 

An additional method involved the use of micrometer-sized magnetic hybrid colloid decorated with AgNPs, considering the preservation of environmental media [86]. In fact, these particles can be easily collected after use with a simple magnet, limiting the potential risks to human health and the environment. Tests in different environmental conditions were conducted against the murine norovirus (MNV), the bacteriophage ΦX174 and the adenovirus (AdV2). The AgNP hybrid colloidal system showed an antiviral effect against the bacteriophage and MNV, depending on the AgNPs’ concentration, size and time of exposure (as already explained in the previous section), but not towards the adenovirus. Strong acidic and alkaline conditions (pH = 2 and pH = 12) reduced the antiviral activity of the AgNP hybrid colloidal system. The authors hypothesized that the adenovirus size, which ranges from 70 to 100 nm, compared to 27-33 nm for bacteriophage ΦX174 and 28–35 nm for MNV, could influence the interaction between the virus and AgNPs and, consequently, alter their antiviral activity.

This type of application has a potential huge impact on human health on a global scale, since it allows the obtaining of pure and disinfected water with systems that do not contaminate or pollute the environment.

#### 3.3.2. Air Filtration Systems

Many microbial agents, such as viruses, bacteria and fungi, can remain in the air for a long time and be transmitted to a susceptible host through inhalation. Moisture in heating, ventilation and air conditioning (HVAC) systems and the accumulation of dust create a favorable environment for bacteria and fungi proliferation [101]. In addition, modern high-efficiency particulate air filters (HEPA) only limit viruses and bacteria transmission, without completely inhibiting the proliferation of bacteria and fungi or inactivating viruses present on the surface or in the filter’s porous structure. This is a potential risk for human health since a possible consequence could be poor air quality and an increase in respiratory diseases. The design of a multilayer filter with different degrees of porosity could be an efficient strategy if the nanoporous membrane is loaded with AgNPs [41]. In this case, the combination of the antimicrobial AgNP properties and the nanoporosity, derived by spun nanofibers, increases the possibility of entrapping and inactivating microorganisms with excellent air purification performance.

As reported in the previous section, an innovative composite coating, already tested towards SARS-CoV-2 on disposable masks [78], was also deposited on metallic and glass fiber air filters for antiviral activity evaluation against three airborne viruses: the rhinovirus (HRV), influenza A virus (IFVA) and respiratory syncytial virus (RSV). This patented coating, composed of a silica matrix embedding silver nanoclusters, was deposited via co-sputtering [102]. The main advantages of this thin layer are the gradual and controlled silver ion release, without nanoparticle dispersion into the surrounding environment, thanks to the silica matrix; the ability to conform adaption to any material, including flexible or thermal-sensitive substrates; and the “green” co-sputtering method used for the deposition, which is easily industrially scaled-up [103,104]. Besides its verified antibacterial properties without altering filtering performance [105], the deposited coating drastically reduced RSV and IFVA titer, while no changes in the number of infective particles were visible in the case of HRV. This probably depends on the different structure of the viruses, as HRV is a more resistant naked virus and less susceptible to different conditions, such as pH or temperature, than viruses endowed with a lipid bilayer, such as RSV and IFVA.

Similarly, an air filter with AgNP-coated silica particles was studied against aerosolized bacteriophage MS2 and IFVA, considering continuous airflow conditions [17,88,106]. A schematic comparison of the two systems is represented in Figure 4. 

This system had several advantages, such as the possibility of avoiding aggregation phenomena and collecting particles after use, and the efficiency of the production process without dangerous effects on human health and on the environment [17]. The increment of the areal density coated by SiO2–Ag and the AgNP amount improved the antiviral effect, as MS2 and the IFVA virus interacted directly with Ag nanoparticles of the filter. However, an increment of the deposition time of the viral particles together with the presence of deposited dust on the filter decreased the filtration quality, limiting the inactivation of the virus. In particular, dust prevents direct contact between Ag nanoparticles and viruses [88]. In fact, after 1 h of exposure to the coated Ag–SiO_2_ filter, 80% of the viral protein HA was damaged and 20% of the activity of the viral protein NA decreased [17].

This type of application has an important impact on everyday life: it is possible to obtain an eco-friendly system capable of monitoring and improving air quality, with great attention to the indoor environment, in order to reduce the risks of human respiratory diseases.

### 3.4. Applications of AgNPs in the Food Industry

AgNPs are generally used in the food industry by being embedded in polymeric matrices to create films or containers for food, in order to avoid contamination with bacteria and fungi [107]. However, some viruses, such as the human norovirus (HNV), HRoV and HAV, can also be transmitted through the fecal–oral route, thus spreading through food and contaminated fomites [108]. To counteract this problem, a polymeric film of poly(3-hydroxybutyrate-co-3 mol%-3-hydroxyvalerate) (PHBV3) with an electrospun coating containing AgNPs, was developed as a food packaging application [57]. The composite Ag/polymer film demonstrated an antiviral activity, increasing proportionally with the Ag amount, towards FCV and MNV, and was used as a predictive model of human viral pathogens. 

Despite these preliminary studies, the use of AgNPs in the food industry must be carefully evaluated case by case. In fact, it is necessary to evaluate any harmful effects on humans, as an accumulation of nanoparticles and the release of Ag ions can damage the liver, spleen, kidneys and lungs, but also the immune system [109,110,111]. In Section 4, the toxicity aspect and limitation of AgNPs will be better explored.

### 3.5. Applications of AgNPs in the Textile Industry

Beyond the recent pandemic, the main demand for antiviral textiles has always been in the healthcare field, where patients and workers are at high risk of contracting viruses [112]. Textiles generally used in hospitals and clinics or in direct contact with patients, such as fabrics for bandages and personal protective equipment (PPE), could be a vehicle for transmission of some resistant viruses. As example, a recent publication indicates that SARS-CoV-2 can persist, in the presence of a soil load, for up to 21 days on experimentally inoculated PPE, including materials from filtering facepiece respirators (N-95 and N-100 masks) and plastic visors. Other main applications for antiviral textiles involve crowded places and environments with a certain risk of cross-infection, such as public transport, kindergarten classrooms and public offices. Moreover, in the context of the COVID-19 pandemic, many textile fashion companies, in particular in Italy, decided to invest in the development of textile products with antiviral protection, along with antibacterial activity, to satisfy consumer demand and need. In particular, Albini Group, an Italian company specializing in high-end fabrics for shirts, developed a new technology called ViroFormula^TM^ [113], ideal for the production of shirts, jackets and trousers, but also masks, gowns and any other garment. This technology was industrialized on the basis of Viroblock technology, formulated by HeiQ, a Swiss company. Viroblock developed a system for conferring antimicrobial and antiviral properties on fabrics, exploiting the synergic action of AgNPs and phospholipid vesicles, which destroys the virus envelope by binding to the virus and preventing its replication [114]. This system was used for the production of protective masks, mattresses, pillows and nonwoven medical garments. Its effectiveness was tested against different types of influenza viruses and remained unchanged after 30 washes at 60 °C. In addition, it was tested against SARS-CoV-2, showing a reduction of 99.99% of the virus titer [114].

In general, the main procedures to confer antiviral properties on textile materials via AgNPs involve the addition of the antiviral agent directly into the polymer solution during the spinning process or through a finishing method as the impregnation of the final product. Multifunctional poly(methylmethacrylate) nanofibers with ZnO nanorods and AgNPs demonstrated antiviral activity against the bovine coronavirus (BCV) and bovine parainfluenzavirus (BPIV3) [38]. The solution containing the polymer ZnO obtained via hydrothermal technique and AgNPs prepared via AgNO_3_ reduction was directly electrospun on mats for producing protective clothing. Another study verified the antiviral performance of AgNPs sprayed on polyester/viscose spunlace wipes for use in surface disinfection [39]. The AgNPs were prepared by means of different processes using reducing agents, such as trisodium citrate with cotton yarn; through an aqueous solution of PVA in the presence of glucose; or via photochemical reaction of polyacrylic acid and silver nitrate solution. All of the proposed methods allowed a good distribution and stability of the nanoparticles, but only the fabric doped with AgNPs obtained using cotton as a reducing agent and from the photochemical reaction process showed an antiviral activity towards MERS-CoV.

## 4. Notes about Limitations of AgNPs

An important aspect to be considered about metallic nanoparticles is their potential toxicity and the effects they could have on human health and the environment. A complete description of AgNP toxic effects is beyond the focus of this review. Briefly, nanoparticles can have cytotoxic effects on animal cells and when administered orally or inhaled they could accumulate in different organs, such as the liver, lungs, spleen and kidneys [115].

Silver in the ionic form is likely more toxic than in the nanoparticle form, but that difference could reflect their different biokinetics. However, AgNPs and ions have a similar pattern of toxicity, probably reflecting that the effect of AgNP is primarily mediated by released ions [116,117]. Several studies regarding the toxicity of silver on human cells revealed different consequences, such as increased inflammatory responses [118] and immunotoxicity [119]. The toxic effects depend on time of exposure, dose [120], NP size [121], shape [121] and presence of surface coating [122,123]. 

On the other hand, the effects of AgNPs on the environment depend on how they are released. In fact, AgNPs can react with chlorides, sulphides and other substances by changing the original properties. Nevertheless, silver in its solid state can be disposed of in landfills, incinerated during heat treatments or thrown into waste or natural waters [108]. It is therefore essential to find a way to recover the nanoparticles in an environmentally safe way; for example, by exploiting the magnetic properties of the AgNP–magnetic hybrid colloid to easily collect the nanoparticles after their use [85]. In order to safeguard the environment, the biosynthesis of nanoparticles following green rather than chemical methods is preferred to avoid the use of chemicals and the related production of waste that must be disposed of. In particular, the production of AgNPs starting from plant extracts was simple and cost-effective and allowed the obtaining of green and stable antiviral agents as alternative therapeutic methods for viral diseases [48]. 

## 5. Conclusions and Future Perspective

From the reported analysis of the most recent literature, we can conclude that AgNPs should have precise features to better exert their antiviral action. 

Considering spherical nanoparticles, which are the most studied ones, the first important parameter is size. In particular, we showed that if the size is small, generally around 10 nm, the antiviral effect is stronger, while an increase in the diameter causes a decrease in the antiviral efficacy. The second factor is the concentration of the nanoparticles in the system: a higher AgNPs concentration leads to a stronger effect. Size and concentration are closely related to each other, since an excessive amount of silver can cause toxicity problems. Another aspect that could influence the AgNP properties is the functionalization or the presence of AgNPs in composite materials. In these cases, the antiviral effect could be emphasized through the use of chemical agents or coatings that are intrinsically antiviral; if the material has no antiviral effect itself, the presence of a matrix could, however, represent a physical impediment for virus interaction with the host cell. In addition, specific types of functionalization can mimic cell receptors, promoting the interaction between functionalized AgNPs and viruses, and preventing virus binding to cells. Finally, the exposure time can also influence the antiviral activity of the system.

At the end of this review, we may also conclude that, thanks to their broad-spectrum antiviral activity, AgNPs can potentially be involved in different fields. Indeed, numerous studies were conducted to develop new antiviral approaches for the prevention and treatment of viral diseases. Nevertheless, a few devices based on AgNPs were actually put on the market, such as face masks or clothes, while other systems, in particular the ones involved in the treatment of infectious diseases, are only prototypes mainly studied in laboratories. In particular, we found that AgNPs have strong potential in the field of antiviral materials, thanks to their virucidal mechanism of action, which as a result is the most represented one. The applications of AgNPs in water and air purification systems and in the textile industry are the most promising and could potentially have a strong impact on life quality. In particular, in a period characterized by the spread of SARS-CoV-2, the development of antiviral equipment, which can purify air or improve surgical mask efficiency, could be a strategic method for returning to normal life. In addition, quality of life can be improved by the application of AgNP-based systems in objects used daily; for example, in food packaging or textiles. These new technologies could also find application in the future against new viruses with pandemic potential or against well-known viruses that are difficult to control. 

In conclusion, further studies should be conducted to clarify the mechanisms of action of AgNPs, to improve their antiviral efficiency and to identify methods that allow the modulating of, and specifically directing, their antiviral activity when AgNPs are applied to different devices. Additionally, further efforts should also be focused on the impacts that silver nanoparticles have on the environment and, in particular, on the production of waste. It would be interesting to find a simple and safe way to collect silver nanoparticles after use or, alternatively, to recycle the devices and involve them in other applications, reducing production costs.

## Figures and Tables

**Figure 1 microorganisms-11-00629-f001:**
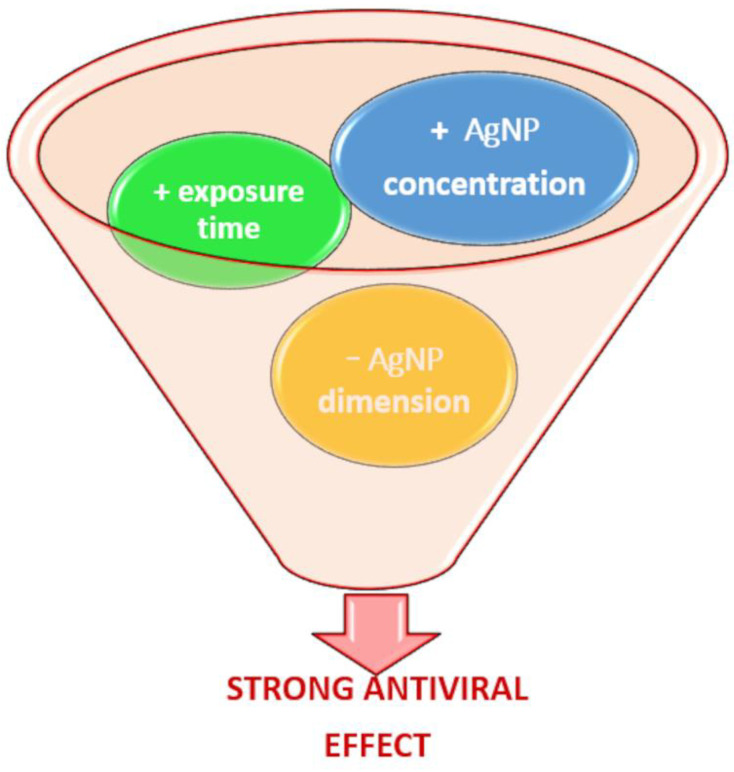
Scheme of the effect of dimension, concentration and exposure time on the antiviral effect of AgNPs. A strong antiviral effect is obtained if AgNPs in small dimensions are used in higher concentrations and exposed to viruses for longer times.

**Figure 2 microorganisms-11-00629-f002:**
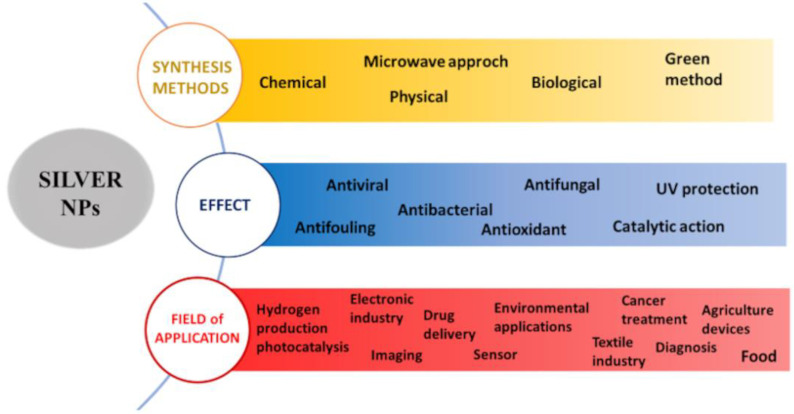
Main synthesis methods, effects and potential applications of AgNPs.

**Figure 3 microorganisms-11-00629-f003:**
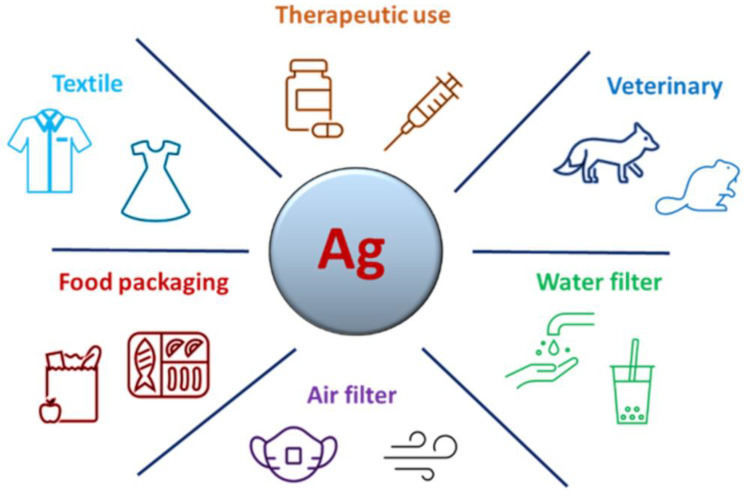
Main potential applications of AgNPs in the virology field.

**Figure 4 microorganisms-11-00629-f004:**
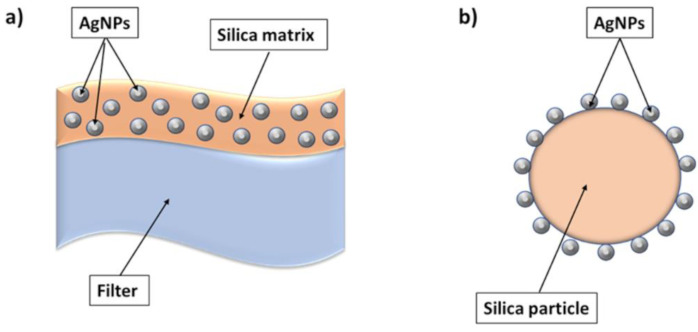
Comparison between two different silver–silica systems: (**a**) nanostructured silica composite coating containing silver nanoparticles; (**b**) silica particles decorated with silver nanoparticles.

**Table 1 microorganisms-11-00629-t001:** Summary of the most relevant studies published since 2010 concerning the antiviral activity of AgNPs.

Virus (and Family)	In Vitro/In Vivo Study	AgNPs Synthesis/Methods	Mechanism of Action	Main Features of and Factors Influencing the Antiviral Activity	Ref.
Influenza virus A (IFVA) (*Orthomyxoviridae family*)	In vitro	AgNPs dispersed into a chitosan matrix	Probable alteration or inactivation of the viral particle	*Shape*: spherical	[21]
				*Size*: 3.5, 6.5, and 12.9 nm. Stronger antiviral effect with smaller AgNPs	
				*Concentration*: 73, 62 and 77 µg/mL. ↑C.→↑A.e.	
				*Modification/functionalization:* chitosan matrix without any antiviral effect, but used only for avoiding NP diffusion into the environment	
				*Exposure time*: /	
	In vitro	New disinfectant formulation combining surfactants and AgNPs	/	*Shape*: spherical	[35]
				*Size*: 13.2 ± 4 nm	
				*Concentration: /*	
				*Modification/functionalization: /*	
				*Exposure time: /*	
	In vitro	AgNPs from Panax ginseng root extract	/	*Shape*: quasi-spherical	[18]
				*Size:* range 5–15 nm	
				*Concentration:* 0.005, 0.01, 0.015, 0.02, 0.025 M. ↑C.→↑A.e.	
				*Modification/functionalization: /*	
				*Exposure time: /*	
	In vitro	AgNP–decorated silica hybrid composite	Damaging of the viral protein HA and NA	*Shape: /*	[17]
				*Size:* 30 nm	
				*Concentration:* 1 × 10^8^, 1 × 10^9^, 5 × 10^9^, 1 × 10^10^ particles/mL. ↑C.→↑A.e.	
				*Modification/functionalization: /*	
				*Exposure time:* 1, 2 3, 6 and 24 h. ↑E.t.→↑A.e	
	In vitro	Oseltamivir-modified AgNPs (Ag@OTV)	Induction of morphologic abnormalities in the viral particle. Inhibition of HA and NA activity. Ag@OTV inhibited the accumulation of ROS by the H1N1 virus	*Shape: /*	[16]
				*Size:* 2–3 nm	
				*Concentration: /*	
				*Modification/functionalization:* Oseltamivir modification. Improvement of antiviral effect and reduction of toxicity towards cells	
				*Exposure time: /*	
	In vitro and in vivo	AgNPs via oxidation-reduction reaction	Destruction of morphologic viral structure	*Shape:* spherical	[20]
				*Size:* 9.5 ± 0.8 nm	
				*Concentration:* 12, 5, 25, 50 µg/mL. ↑C.→↑A.e.	
				*Modification/functionalization: /*	
				*Exposure time*: 30, 60 and 120 min. Alteration of viral morphological structure with higher exposure time	
	In vitro	AgNPs (unspecified synthesis)	/	*Shape: /*	[19]
				*Size*: range 5–20 nm	
				*Concentration*: 6, 25, 12, 5, 25, 50, 100, 200 µg/mL. Better antiviral effect with AgNP concentration of 50 µg/mL	
				*Modification/functionalization: /*	
				*Exposure time*: 72 and 96 h. ↑E.t.→↑A.e.	
Transmissible gastroenteritis virus (TGEV) (*Coronaviridae* *family)*	In vitro	AgNPs, Ag nanowires, Ag colloids (unspecified synthesis)	/	*Shape*: Nanowires, spherical nanoparticles and colloids	[30]
				*Size:* NPs less than 20 nm, nanowire diameter of 60 nm and 400 nm, colloids about 10 nm	
				*Concentration*: 3.125, 6.25 and 12.5 µg/mL. ↑C.→↑A.e.	
				*Modification/functionalization:* Polyvinylpyrrolidone (PVP) as capping agent. Higher antiviral effect with nanowires and nanoparticles; the presence of PVP coating in Ag colloids decreases their antiviral activity	
				*Exposure time: /*	
Severe acute respiratory syndrome coronavirus virus 2 (SARS-CoV-2) (*Coronaviridae family)*	In vitro	AgNPs (commercial)	Inhibition of extracellular viral particles	*Shape: /*	[36]
				*Size*: 2, 15, 50, 80, 100 nm. Stronger antiviral effect with 2 and 15 nm dimension	
				*Concentration*: 0, 1, 0.5, 1, 2, 5 and 10 ppm. Higher concentration could become cytotoxic	
				*Modification/functionalization: /*	
				*Exposure time: /*	
	In vitro and in a prospectiverandomized study of 231 participants	AgNPs (Argovit^TM^)	/	*Shape:* spherical	[37]
				*Size:* 35 ± 15 nm	
				*Concentration*: /	
				*Modification/functionalization: /*	
				*Exposure time:* 24, 48 and 72 h	
Bovine coronavirus (BCoV) (*Coronaviridae family)*	In vitro	poly(methyl methacrylate) (PMMA) nanofibers decorated with ZnO nanorods and AgNPs (PMMA/ZnO−Ag NFs)	/	*Shape: /*	[38]
				*Size:* 510 ± 122 nm	
				*Concentration: /*	
				*Modification/functionalization:* System of poly(methylmethacrylate) (PMMA) nanofibers with ZnO nanorods and AgNPs	
				*Exposure time:* 1 h and 24 h. ↑E.t.→↑A.e.	
Middle East respiratory syndrome coronavirus (MERS-CoV) and severe acute respiratory syndrome coronavirus (SARS-CoV) (*Coronaviridae* *family)*	In vitro	AgNPs prepared with different methods	/	*Shape: spherical*	[39]
				*Size*: 7.1 ± 2.4 nm	
				*Concentration*: 3.13–100 ppm. Stronger antiviral activity at a concentration of AgNPs 3.13 ppm. Higher concentrations are toxic	
				*Modification/functionalization: /*	
				*Exposure time*: 24, 48 and 72 h. ↑E.t.→↑A.e.	
Feline coronavirus (FCoV) (*Coronaviridae* *family)*	In vitro	AgNP-anchored Graphene Oxide (GO-Ag)	/	*Shape:* spherical	[40]
				*Size:* 5–25 nm	
				*Concentration:* 0.1, 1, 10, 100 mg/mL. Higher antiviral activity with higher AgNP concentration	
				*Modification/functionalization:* AgNPs attached on graphene oxide. GO acts as stabilizing and supporting agent	
				*Exposure time: /*	
Porcine deltacoronavirus (PDCoV) (*Coronaviridae* *family)*	In vitro	PP nonwoven substrate with a layer of PA6 electrospun nanofiber, impregnated with AgNPs	/	*Shape: /*	[41]
				*Size: /*	
				*Concentration: /*	
				*Modification/functionalization:* PA6 nanofiber impregnated with AgNPs	
				*Exposure time*: 15, 30 and 60 min. ↑E.t.→↑A.e.	
Poliovirus (*Picornaviridae family)*	In vitro	AgNPs synthesized via electrochemical method	/	*Shape*: /	[42]
				*Size*: 7.1 ± 2.4 nm	
				*Concentration*: 3.13–100 ppm. Stronger antiviral activity at AgNP concentration of 3.13 ppm. Higher concentrations are toxic	
				*Modification/functionalization: /*	
				*Exposure time*: 24, 48 and 72 h. ↑E.t.→↑A.e.	
Hepatitis A virus (HAV) and Coxakievirus B4 (CoxB4) (*Picornaviridae family)*	In vitro	AgNP synthesis using *Lampranthus coccineus* and *Malephora lutea* aqueous and hexane extracts	/	*Shape*: spherical	[23]
				*Size*: range between 10.12–27.89 nm for *L. coccineus* and 8.91–14.48 nm for *M. Iutea*	
				*Concentration*: /	
				*Modification/functionalization: /*	
				*Exposure time: /*	
Human immunodeficiency virus (HIV) (*Retroviridaefamily)*	In vitro	AgNPs into foamy carbon (1–10 nm) AgNPs coated by poly (N-vinyl-2-pyrrolidone) (PVP) AgNPs modified by bovine serum albumin (BSA)	The physical interaction of AgNPs with the viral particle probably inhibits virus binding to the host cell	*Shape*: /	[14]
				*Size*: 16.19 ± 8.68 nm, 6.53 ± 2.41 nm, 3.12 ± 2.00 nm, depending on the preparation method	
				*Concentration*: /	
				*Modification/functionalization:* AgNP-coated foamy carbon, PVP-coated AgNPs, BSA- functionalized AgNPs. Reduction of antiviral effect for the presence of PVP and BSA	
				*Exposure time*: /	
	In vitro	AgNPs synthesized with sericin	/	*Shape*: /	[43]
				*Size*: 3.78 ± 1.14 nm (at optimized sericin concentration)	
				*Concentration*: /	
				*Modification/functionalization: /*	
				*Exposure time*: /	
Herpes simplex virus type 1 or 2 (HSV-1, HSV-2) (*Herpesviridae* *family)*	In vitro	AgNPs capped with mercaptoethane sulfonate group (Ag–MES)	Suggested inhibition of virus binding to cell	*Shape*: /	[26]
				*Size*: 4 ± 1 nm	
				*Concentration*: 200, 400 and 800 µg/mL. ↑C.→↑A.e. (infection is blocked at a concentration of 400 µg/mL)	
				*Modification/functionalization:* mercaptoethane (MES) sulfonate group. MES, which decorated AgNPs, acts as a multivalent inhibitor since it mimics HS group on the membrane of host cell, inhibiting the virus	
				*Exposure time*: /	
	In vitro and in vivo	Tannic Acid (TA)- Modified AgNPs	TA-AgNPs bind the virus, blocking cell infection; TA-AgNPs elicited B cells’ activation and plasma cells’ homing	*Shape*: /	[44]
				*Size*: 33 nm	
				*Concentration*: /	
				*Modification/functionalization:* Tannic acid as additional adjuvant for enhancing antiviral properties	
				*Exposure time: /*	
	In vitro and in vivo		Tannic acid-modified AgNPs block virus attachment and entry by direct interaction	*Shape*: /	[24]
				*Size*: 13 ± 5 nm, 33 ± 7 nm, 46 ± 9 nm and 10 ± 5 nm. Higher antiviral activity with 33 nm nanoparticles	
				*Concentration*: 0.5, 1, 2.5 µg/mL. The effect of the concentration depends on AgNPs’ size	
				*Modification/functionalization:* Tannic acid as additional adjuvant for enhancing antiviral properties	
				*Exposure time*: 60 min	
	In vitro	AgNP synthesis using *Lampranthus coccineus* and *Malephora lutea* aqueous and hexane extracts	Probable binding of AgNPs to viral envelope glycoproteins, preventing viral penetration into the host cell	*Shape*: spherical	[23]
				*Size*: range between 10.12–27.89 nm for *L. coccineus* and 8.91–14.48 nm for *M. Iutea*	
				*Concentration*: /	
				*Modification/functionalization: /*	
				*Exposure time: /*	
	In vitro and in vivo	Tannic Acid–AgNPs into Carbopol 974P gel	Inhibition of viral attachment, penetration and cell-to-cell transmission; Direct interaction of NPs with viral envelope or its proteins	*Shape*: /	[25]
				*Size*: 33 ±13 nm	
				*Concentration*: 25 or 50 ppm	
				*Modification/functionalization:* Tannic acid as additional adjuvant for enhancing antiviral properties. Presence of Carbopol 974P hydrogels	
				*Exposure time:* 24 h	
	In vitro	AgNPs synthesized from different types of fungi	/	*Shape*: spherical	[22]
				*Size*: range between 4 and 31 nm. Smaller size, higher inhibition of virus infectivity	
				*Concentration*: 0.1, 0.5, 1, 5 and 10 µg/mL. ↑C.→↑A.e., depending on fungal species	
				*Modification/functionalization: /*	
				*Exposure time* *: /*	
Respiratory syncytial virus (RSV) (*Paramyxoviridae* *family*)	In vitro and in vivo	PVP-coated AgNPs (commercial)	AgNP-treated and RSV-infected mice showed reductions in pro-inflammatory cytokines and chemokines	*Shape*: spherical	[45]
				*Size:* range 8–12 nm	
				*Concentration*: 0, 10, 25, 50 µg/mL. ↑C.→↑A.e.	
				*Modification/functionalization:* PVP coating	
				*Exposure time*: /	
	In vitro	Curcumin-modified silver nanoparticles (cAgNPs)	Direct virus inactivation	*Shape*: /	[46]
				*Size:* 11.95 ± 0.23 nm	
				*Concentration*: 0.008, 0.015, 0.03, 0.06, 0.12, 0.24 nM. Higher virus titer reduction with concentration between 0.008 and 0.06 nM.	
				*Modification/functionalization:* AgNPs modified by curcumin	
				*Exposure time*: 1 h	
Peste des petits ruminants virus (PPRV) (*Paramyxoviridae* *family*)	In vitro	AgNP synthesis from silver nitrate using extract of the *Argemone maxicana* leaf as reducing agent	Inhibition of virus entry	*Shape*: /	[47]
				*Size*: 20 nm	
				*Concentration*: range 1.23–900 µg/mL. ↑C.→↑A.e.	
				*Modification/functionalization: /*	
				*Exposure time*: 90 min	
Parainfluenza virus (HPVI) (*Paramyxoviridae family*)	In vitro	AgNP synthesis from different types of fungi	/	*Shape*: spherical	[22]
				*Size*: range between 4 and 31 nm. Smaller size, higher inhibition of virus infectivity	
				*Concentration*: 0.1, 0.5, 1, 5 and 10 µg/mL. ↑C.→↑A.e., depending on fungal species	
				*Modification/functionalization: /*	
				*Exposure time* *: /*	
Bovine parainfluenza virus 3 (BPIV3) (*Paramyxoviridae family*)	In vitro	Poly(methyl methacrylate) (PMMA) nanofibers decorated with ZnO nanorods and Ag nanoparticles (PMMA/ZnO−Ag NFs)	/	*Shape: /*	[38]
				*Size:* 510 ± 122 nm	
				*Concentration: /*	
				*Modification/functionalization:* System of poly(methylmethacrylate) (PMMA) nanofibers with ZnO nanorods and AgNPs	
				*Exposure time:* 1 h and 24 h. Higher antiviral activity with higher exposure time	
Chikungunya virus (CHIKV) *(Togaviridae* *family)*	In vitro	AgNP synthesis via plants with bioactive phytoconstituents linked to them	/	*Shape*: spherical	[48]
				*Size*: range 50–120 nm depending on the plant	
				*Concentration: /*	
				*Modification/functionalization: /*	
				*Exposure time*: /	
	In vitro	AgNP synthesis using *P. guajava* leaves extract	/	*Shape: /*	[49]
				*Size:* 75–99 nm	
				*Concentration: /*	
				*Surface functionalization: /*	
				*Exposure time* *: /*	
Hepatitis C virus (HCV) (*Flaviviridae* *family)*	In vitro	AgNPs from total extract and petroleum ether fraction of *Amphimedon*	Anti–HCV NS3 helicase and protease activity	*Shape: spherical*	[50]
				*Size:* 8.22–14.30 nm and 8.22–9.97 nm	
				*Concentration: /*	
				*Modification/functionalization: /*	
				*Exposure time* *: /*	
Dengue virus (DENV) (*Flaviviridae* *family)*	In vitro	AgNP synthesis using *Centroceras clavulatum* aqueous extract	/	*Shape*: spheric and cubic	[51]
				*Size:* 35–65 nm	
				*Concentration*: 6.25, 12.5, 25 and 50 µg/mL. ↑C.→↑A.e.	
				*Modification/functionalization: /*	
				*Exposure time*: 48 h	
	In vitro	*B. cylindrica*-synthesized AgNPs	/	*Shape*: spherical	[52]
				*Size:* 30–70 nm	
				*Concentration*: 10, 20 and 30 µg/mL. ↑C.→↑A.e.	
				*Modification/functionalization: /*	
				*Exposure time*: 24 h	
	In vitro	*M. oleifera*-synthesized AgNPs	/	*Shape*: spherical	[53]
				*Size*: 100 nm	
				*Concentration*: /	
				*Modification/functionalization: /*	
				*Exposure time*: /	
Resus rotavirus (RRV) (*Reoviridae* *family)*	In vitro	AgNP synthesis using NaBH4 into AgNO3 solution containing sodium citrate	In vivo upregulation of TGF-β	*Shape*: /	[54]
				*Size*: 10 ± 5 nm	
				*Concentration*: 0, 1, 5, 10 and 20 µg/mL. ↑C.→↑A.e.	
				*Modification/functionalization: /*	
				*Exposure time*: /	
T4 bacteriophage (*Myoviridae* *family)*	In vitro	Granular activated carbon (GAC) modified with silver and/or copper oxide nanoparticles	Suggested inactivation of the viral particle	*Shape*: /	[55]
				*Size*: 25–40 nm	
				*Concentration*: 0.5% and 1% *w*/*w*	
				*Modification/functionalization:* Presence of GAC as second element	
				*Exposure time*: /	
Rift Valley fever virus (RVFV) *(Phenuiviridae* *family)*	In vitro and in vivo	AgNPs functionalized with PVP (Argovit^TM^)	/	*Shape*: spherical	[56]
				*Size*: 35 ± 15 nm	
				*Concentration*: 1.5–12 µg/mL. ↑C.→↑A.e.	
				*Modification/functionalization:* poly(vinylpyrrolidone) (PVP)	
				*Exposure time: /*	
Feline calicivirus (FCV) (*Calicidiviridae* *family)*	In vitro	AgNPs	Alteration of the viral capsid protein	*Shape:* spherical	[27]
				*Size*: 10, 75, 110 nm. Stronger effect with smaller dimension	
				*Concentration*: 25, 50, 100 µg/mL. ↑C.→↑A.e.	
				*Modification/functionalization: /*	
				*Exposure time*: 15 min, 30 min, 1 h, 2 h, 4 h	
Murine norovirus (MNV) and Feline Calicivirus (FCV) (*Calicidiviridae* *family)*	In vitro	Ag nitrate, AgNPs, poly (3- hydroxybutyrate-co-3 mol%-3-hydroxyvalerate) (PHBV3) coated with PHBV18/AgNP fiber	Virucidal activity of the AgNP film	*Shape: /*	[57]
				*Size: AgNPs 7* ± 3 nm; PHBV18/AgNP fibers 1.1 ± 0.40 µm	
				*Concentration:* For AgNPs 0, 2.1, 10.5 and 21 mg/L. ↑C.→↑A.e.	
				*Modification/functionalization:* PHBV18/AgNP fibers electrospun on PHBV3	
				*Exposure time:* 0, 30, 75 and 150 days: for Ag nitrate, ↑E.t.→↑A.e., until 75 days, with 150 days antiviral activity was reduced. For AgNPs, higher antiviral activity with higher exposure time. For PHBV3/PHBV18/Ag NPs: 24 h	
Vaccinia virus (VACV) (*Poxvirus* *family)*	In vitro	AgNPs (commercial)	Inhibition of viral entry through a macropinocytosis-dependent mechanism	*Shape*: /	[58]
				*Size*: 25 nm ± 10 nm	
				*Concentration*: 32 µg/mL	
				*Modification/functionalization: /*	
				*Exposure time*: 1 h	
Adenovirus (ADV) *(Adenoviridae* *family)*	In vitro	AgNPs by chemical redox method using tannic acid	Damaging of the viral structure	*Shape:* spherical	[59]
				*Size*: 10, 75, 110 nm. Stronger effect with smaller dimension	
				*Concentration*: 25, 50, 100 µg/mL. ↑C.→↑A.e.	
				*Modification/functionalization: /*	
				*Exposure time*: 48 and 96 h.	
Hepatitis B virus (HBV) (*Hepadnaviridae family)*	In vitro	AgNPs synthesized in HEPES buffer from silver nitrate	AgNPs binding to HBV double-stranded DNA or to the viral particle	*Shape:* /	[60]
				*Size*: 10, 50, 800 nm. Good antiviral effect of AgNPs of 10 and 50 nm. AgNPs of 800 nm induce cytotoxic effect	
				*Concentration*: 5–50 µM (for AgNPs of 10 and 50 nm). ↑C.→↑A.e.	
				*Modification/functionalization: /*	
				*Exposure time*: 0–96 h. ↑E.t.→↑A.e.	
Bunyamwera virus (BUNV) (*Peribunyaviridae family)*	In vitro	AgNPs (commercial)	Alteration of virus morphology and alteration of replication organelles (RO)	*Shape: /*	[61]
				*Size*: 10 nm	
				*Concentration*: 1.5, 2.4, 6, 12 and 24 µg/mL. ↑C.→↑A.e.	
				*Modification/functionalization: /*	
				*Exposure time*: 10 h	
Infectious bursal disease virus (IBDV) (*Birnaviridae* *family)*	In vitro	AgNP-anchored Graphene Oxide (GO-Ag)	/	*Shape:* spherical	[40]
				*Size:* 5–25 nm	
				*Concentration:* 0.1, 1, 10, 100 mg/mL. ↑C.→↑A.e.	
				*Modification/functionalization:* AgNPs attached on graphene oxide. GO acts as stabilizing and supporting agent	
				*Exposure time: /*	
	In vitro	AgNPs	/	*Shape: /*	[62]
				*Size:* range 50–59 nm	
				*Concentration:* 0, 10, 20, 50 ppm. Higher antiviral effect with a dosage of 20 ppm	
				*Modification/functionalization: /*	
				*Exposure time: /*	

AgNPs = silver nanoparticles; C. = concentration; A.e. = Antiviral effect; E.t. = Exposure time; ↑ = increase. / = data not available in literature.

**Table 2 microorganisms-11-00629-t002:** Summary of the mechanisms of action of AgNPs against different viruses.

	Virucidal Activity	Inhibition of the Early Steps of Replication (Binding Inhibition, Entry Inhibition)	Inhibition of the Late Steps of Replication (Viral Replication Inhibition or Viral Enzyme Inhibition)
**Virus**	IFV-A, SARS-CoV-2, RSV, FCV, ADV, BUNV, MNV, T4 bacteriophage	HIV, HSV-1, HSV-2, PPRV, VACV	HCV, HBV

IFV-A: influenza virus type A; SARS-CoV-2: severe acute respiratory syndrome coronavirus 2; RSV: respiratory syncytial virus; FCV: feline calicivirus; ADV: adenovirus; BUNV: Bunyamwera virus; MNV: murine norovirus; T4 bacteriophage; HIV: human immunodeficiency virus; HSV-1: herpes simplex virus type 1; HSV-2: herpes simplex virus type 2; PPRV: peste des petits ruminants virus; VACV: vaccinia virus; HCV: hepatitis C virus; HBV: hepatitis B virus.

**Table 3 microorganisms-11-00629-t003:** Summary of the main applications and uses of AgNPs as antiviral agents.

Field	Description of Application	System	Tested Virus	Study Level	Ref.
Health sector	Potential therapeutic use	AgNPs	IFVA (H3N2)	Laboratory study	[20]
	Potential therapeutic use	Tannic acid-modified AgNPs (TA-AgNP)	HSV-2	Laboratory study	[24]
	Potential therapeutic use	TA-AgNPs into Carbopol 974P gel	HSV-1 and HSV-2	Laboratory study	[25]
	Potential therapeutic use	TA-AgNPs	HSV-2	Laboratory study	[44]
	Potential therapeutic use	AgNPs	RSV	Laboratory study	[45]
	Potential therapeutic use	AgNPs functionalized with PVP (Argovit^TM^)	RVFV	Prototype	[56]
	Prevention	AgNPs (Argovit^TM^)	SARS-CoV-2	Prototype	[37]
	Coating for surfaces	Protective hybrid coatings with AgNPs realized by means of sol gel	HIV-1, DENV, HSV-1, IFVA, CoxB3	Prototype	[76]
	Coating for condoms for the prevention of sexually transmitted viruses	AgNP coating by immersion	HIV, HSV-1, HSV-2	Prototype	[77]
	Surgical mask doped with AgNPs by immersion	AgNPs obtained by electrochemical method and incorporating an aqueous solution	IFVA (H5N1)	Prototype	[35]
	Co-sputtered disposable mask	Silver nanoclusters/silica composite coating by co-sputtering method	SARS-CoV-2	Prototype	[78]
	Part of respiratory and surgical masks	Graphene–silver nanocomposite	FCoV, IBDV	Prototype	[40]
Veterinary sector	Potential therapeutic use	AgNPs synthesized by chemical reduction method	IBDV	Laboratory study	[62]
	Potential therapeutic use	Biologically synthesized AgNPs	PPRV	Laboratory study	[47]
	Potential therapeutic use	AgNPs	RVFV	Laboratory study	[56]
	Potential therapeutic use	AgNPs	CVD	Laboratory study	[79]
	Potential therapeutic use	AgNPs	WSSV	Laboratory study	[80,81]
	Potential therapeutic use	AgNPs from aqueous extracts of clove	NDV	Laboratory study	[82]
Water and air filtration systems	Water treatment	AgNPs produced via *Lactobacillus fermentum* (Biogenic Ag^0^)	UZ1 (bacteriophage), MNV-1	Prototype	[83]
	Water treatment under UV radiation	AgNPs via photochemical reduction of silver nitrate on Aeroxide TiO2 P25 and Anatase TiO2	MS2 (bacteriophage)	Laboratory study	[84]
	Water treatment	Fe_2_O_3_/Ag NPs coating on fiber glass	MS2 (bacteriophage)	Laboratory study	[85]
	Water treatment	AgNP-doped and Ag/Cu NP-doped activated carbon by impregnation	T4 (bacteriophage)	Laboratory study	[55]
	Water treatment	Magnetic hybrid colloid-AgNPs	MNV, ɸX174, AdV2	Laboratory study	[86]
	Water treatment	Colloidal and immobilized Ag nanoparticles on a glass substrate	MS2 and T4 (bacteriophages)	Laboratory study	[87]
	Air filtration	PP nonwoven substrate with a layer of PA6 electrospun nanofiber, impregnated with AgNPs	PDCoV	Prototype	[41]
	Air filtration	Silver nanoclusters/silica composite coating on air filter	IFVA, RSV	Laboratory study	[63]
	Air filtration	Silver nanoparticle-coated silica particle	IFVA, MS2 (bacteriophage)	Laboratory study/Prototype	[17,63,88]
	Air filtration	Nano-Ag^0^/titania-chitosan	MS2 (bacteriophage)	Laboratory study	[89]
Food packaging	Packaging	Polymeric film–AgNPs	FCV and Murine Norovirus (MNV)	Prototype	[57]
Textile industry	Antiviral clothing	AgNPs and phospholipid vesicles in the Viroblock/ViroFormula^TM^	FluVA, SARS-Cov 2	Commercial product	[90]
	Antiviral clothing	Electrospun nanofibers with ZnO Nanorods and Ag NPs	BCV, Bovine Parainfluenza Virus Type 3 (BPIV3)	Laboratory study	[37]
	Disinfectant for polyester/viscose spunlace wipes for use in surface disinfection	AgNPs prepared by reducing agent or aqueous solution of PVA in the presence of glucose or photochemical reaction	MERS-CoV	Prototype	[39]

## Data Availability

Publicly available datasets were analyzed in this study. This data can be found here: https://pubmed.ncbi.nlm.nih.gov/, https://www.scopus.com and https://www.embase.com.

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
