# Peer review of "Silver Nanoparticles: Review of Antiviral Properties, Mechanism of Action and Applications"

_microorganisms, 2023, doi:10.3390/microorganisms11030629_

Round 1
Reviewer 1 Report
Luceri et al. provided an overview of the antiviral properties and mechanism of action of silver nanoparticles and explored how the size, shape, functionalization, and concentration of these particles can influence their effect on viruses. They discussed the potential areas for the application of silver nanoparticles to antiviral drugs and strategies, including biomedical, environmental, and food/textile industries. For each potential application, it is noted whether the device is a laboratory study or a commercial product.
This manuscript is clear and unambiguous. The introduction includes enough details for the reader to be brought into the related topics. Across the main paragraphs, the authors could list lots of published results and give details summaries of each paper to constitute the structure of their review. It should be credited that the authors used a paragraph to stress the limits of the silver nanoparticles, helping the readers learn this subject comprehensively. The conclusion and future perspective part indicated the future directions of the study of silver nanoparticles.
There are some unclear material sciences, including some content of paragraphs 2.3.2 and 2.3.3 and many nouns in paragraph 3. Application field of AgNPs, for example, microwave approach in line 438, sol-gel method in line 486, silica matrix in line 514, colloid decorated with AgNPs in line 637, AgNPs hybrid colloidal system in line 641, spun nanofibers in line 663, a silica matrix embedding silver nanoclusters in line 669, co-sputtering in line 670, Viroblock technology in line 732, etc.
Overall, it is a good paper. However, there are still some issues in this manuscript that need to be addressed:
1. The 11-page Table 1 may be compressed for the sake of readers’ convenient use: the 1st column “Virus family” may be omitted or integrated with the 2nd column “Virus” so that the whole table will be much more neat, for example, in pages 3, 4, 5, 8, and 11 the majority of the 1st and/or 2nd column are empty; same sentences such as "higher antiviral effect with higher (AgNPs) concentration" were used for 19 times, “higher antiviral activity (effect) with higher exposure time” were used for 6 times, perhaps the author can replace these repeated words with specific symbols so that much space can be saved.
2. Figure 1 virtually consists of text, so it may be replaced with a table; otherwise, a graphical figure should be efficient to help researchers in the field of molecular cell biology to better understand it and the related body text.
3. The title of figure 3 is “Main potential applications of AgNPs” while it actually also contains the synthesis methods and effects.
Some other minor comments:
1. In Table 1, at bottom of page 3, “Higher effect with higher concentration” should be “Higher effect with higher exposure time”;
2. In line 94, after “we found” the authors should add “it’s reported”, otherwise readers may think those are the experimental results of the authors of this paper;
3. It is difficult for me to understand the sentence in line 220-222;
4. In line 324, “founding” should be “finding”;
5. In line 373, “w” should be “with”;
6. Line 447, “hogh” should be “high”;
7. Line 467, “probable” should be “probably”;
8. Reference No. 60, no journal was specified;
Reviewer 2 Report
I would like to thank for the opportunity to review this paper.
This paper reviewed the antiviral activity of silver nanoparticles (AgNPs) and talked about its application in different fields. The authors summarized the antiviral activity of AgNPs against a variety of viruses and involved mechanisms, especially the impact of nanoparticles size, shape, concentration, and functionalization on the antiviral activity. They also discussed the application of AgNPs in biomedical applications, considering both human and animal health, environmental ap-21 plications, such as air filtration and water treatment, and the food and textile industry purposes. This is a comprehensive review about AgNPs.
Minor comments:
Please check the information in table 1 and 2 carefully. I found some errors such as page9: in ref 46, the size should be 11.95 ± 0.23 nm; page 11, the in ref 52, the size should be “30-70 nm”, not “30-70 mm”. In addition, I noticed that in table 1 and 2, a lot of information is missing (/), is that not available in the references?
Reviewer 3 Report
A review entitled “Silver Nanoparticles: Review of Antiviral Properties, Mechanism of action and Applications” by Luceri et al. describes the activity of silver nanoparticles against various viruses in addition to the possibility to incorporate them into various biomedical applications to control the spread of viral infections. The review contains promising collecting data and it needs minor revision before accepting to publication in a microorganisms journal.
1- Tables should be containing table footnotes to describe the abbreviation and symbols in each table.
2- The abbreviation should be mentioned completely the first time followed by written as abbreviated. Please check and revise the manuscript.
3- The authors should clarify the methods for AgNPs synthesis in a separate section after the introduction. I recommend the following references for cite: https://doi.org/10.3390/catal12050462; https://doi.org/10.1007/s12011-020-02138-3; https://doi.org/10.3390/jof8040396
4- Figure 2 does not give accurate data, please replace it with a new one to be clearer.
5- I recommend summarizing section 3.2 in tables.
6- The title of figure 4 is incorrect. This figure explains various applications of AgNPs not only in the virology field. Please correct.
7- I recommend the authors add a histogram showing various antiviral mechanisms.
